# Dyslipidemia in Children Treated with a BRAF Inhibitor for Low-Grade Gliomas: A New Side Effect?

**DOI:** 10.3390/cancers14112693

**Published:** 2022-05-29

**Authors:** Marco Crocco, Antonio Verrico, Claudia Milanaccio, Gianluca Piccolo, Patrizia De Marco, Gabriele Gaggero, Valentina Iurilli, Sonia Di Profio, Federica Malerba, Marta Panciroli, Paolo Giordano, Maria Grazia Calevo, Emilio Casalini, Natascia Di Iorgi, Maria Luisa Garrè

**Affiliations:** 1Neuroncology Unit, IRCCS Istituto Giannina Gaslini, 16147 Genova, Italy; antonioverrico@gaslini.org (A.V.); claudiamilanaccio@gaslini.org (C.M.); gianlucapiccolo@gaslini.org (G.P.); mluisagarre@gaslini.org (M.L.G.); 2Department of Neuroscience, Rehabilitation, Ophthalmology, Genetics, Maternal and Child Health, University of Genova, 16132 Genova, Italy; federica-malerba@libero.it (F.M.); marta.panciroli@gmail.com (M.P.); plo.grd@gmail.com (P.G.); emiliocasalini@gaslini.org (E.C.); natasciadiiorgi@gaslini.org (N.D.I.); 3UOC Genetica Medica, IRCCS Istituto Giannina Gaslini, 16147 Genova, Italy; patriziademarco@gaslini.org; 4Department of Clinical Pathology, IRCCS Ospedale Policlinico San Martino, 16132 Genova, Italy; gabriele.gaggero@hsanmartino.it; 5Pharmacy Unit, IRCCS Istituto Giannina Gaslini, 16147 Genova, Italy; valentinaiurilli@gaslini.org; 6Clinical Psychology, IRCCS Istituto Giannina Gaslini, 16147 Genova, Italy; soniadiprofio@gaslini.org; 7Epidemiology and Biostatistics Unit, IRCCS Istituto Giannina Gaslini, 16147 Genova, Italy; mariagraziacalevo@gaslini.org; 8Department of Pediatrics, IRCCS Istituto Giannina Gaslini, 16147 Genova, Italy

**Keywords:** BRAF, vemurafenib, lipid metabolism, cholesterol, triglycerides, dyslipidemia, brain tumor, low grade glioma

## Abstract

**Simple Summary:**

The use of targeted therapies is revolutionizing the prognosis of many cancers; however, there is still limited knowledge of their side effects. Dyslipidemia is often present in cancer patients due to mechanisms that are directly or indirectly related to cancer or therapies. The aim of our study is to investigate the effects of vemurafenib on lipid metabolism in a cohort of pediatric patients treated for brain tumors. For the first time, we describe dyslipidemia as a possible side effect of the BRAF inhibitors. A better understanding of the pathways that are involved in dyslipidemia could also help with a better understanding of the drug-resistance mechanisms in cancer cells.

**Abstract:**

BRAF inhibitors, in recent years, have played a central role in the disease control of unresectable BRAF-mutated pediatric low-grade gliomas (LGGs). The aim of the study was to investigate the acute and long-term effects of vemurafenib on the lipid metabolism in children treated for an LGG. In our cohort, children treated with vemurafenib (*n* = 6) exhibited alterations in lipid metabolism a few weeks after starting, as was demonstrated after 1 month (*n* = 4) by the high plasma levels of the total cholesterol (TC = 221.5 ± 42.1 mg/dL), triglycerides (TG = 107.8 ± 44.4 mg/dL), and low-density lipoprotein (LDL = 139.5 ± 51.5 mg/dL). Despite dietary recommendations, the dyslipidemia persisted over time. The mean lipid levels of the TC (222.3 ± 34.7 mg/dL), TG (134.8 ± 83.6 mg/dL), and LDL (139.8 ± 46.9 mg/dL) were confirmed abnormal at the last follow-up (45 ± 27 months, *n* = 6). Vemurafenib could be associated with an increased risk of dyslipidemia. An accurate screening strategy in new clinical trials, and a multidisciplinary team, are required for the optimal management of unexpected adverse events, including dyslipidemia.

## 1. Introduction

Low-grade gliomas are the most common central-nervous-system (CNS) tumors among children [1]. The prognosis for these tumors is generally excellent, with the 10-year overall survival (OS) between 85 and 95% [2]. In pediatric low-grade gliomas (pLGGs), several factors, such as grading, location, age at diagnosis, the extent of surgery, and biological characteristics, influence the OS. A complete surgical resection, whenever feasible, tends to be curative, and a wait-and-see approach can be adopted. pLGGs that cannot be resected without the risk of permanently damaging important functions, or that progress after resection, require additional therapy.

Chemotherapy can control tumors by up to 50% of pLGGs, and over 90% of patients are still alive 20 years after radiotherapy [3]. Considering the excellent OS, the goal of the medical therapy should be to ensure the stability of the disease and to minimize the early and late effects of therapy. In the last decade, biological drugs have filled an important void in the therapeutic armamentarium against pLGGs when they are not resectable and are refractory or recurrent following standard chemotherapy regimens. In these cases, treatment with biological/targeted therapy has the potential to prevent morbidities and sequelae that are typically associated with chemo-radiotherapy, such as neurocognitive impairment, ototoxicity, nephrotoxicity, peripheral neuropathy, and hypothalamic–pituitary disfunction [4,5,6].

The understanding of the biological pathways that lead to pLGGs has allowed the development of new biological therapies that directly interrupt these pathways. pLGGs most frequently have somatic driver genetic alterations that converge on the activation of the *Rapidly Accelerated Fibrosarcoma/Mytogenic-Activated Protein Kinase* (*RAS/MAPK*) pathway [7]. Rearrangements that afflict the genes *v-Raf murine sarcoma viral oncogene homolog B1*
*(BRAF)* and *KIAA1549–BRAF* are the most frequent somatic driver alterations across all pLGGs [8,9]. The *BRAF V600E* mutation is reported in 20–35% of pLGGs [10,11], and, in particular, it is found in 9% of pilocytic astrocytomas, 50% of gangliogliomas, and 66% of pleomorphic xanthoastrocytomas [8,10]. The *BRAF V600E* mutation confers a worse prognosis because of the insensitivity to traditional chemotherapy, and a higher propensity toward malignant transformation in combination with *CDKN2A* deletion [12]. Hence, inhibiting MAPK signaling by using small-molecule inhibitors, such as BRAF inhibitors (BRAFi), may be a promising strategy in pLGGs [13,14,15,16]. It is also known that the various combinations of MEKi and BRAFi can delay or prevent the occurrence of resistance to a single therapy and may reduce side effects (SEs). [17,18,19,20]. These therapies may lead to a rapid and prolonged response of the tumors [6,19,21,22]. As observed in other gliomas, such as subependymal giant cell astrocytoma (SEGA), in which stopping the mammalian target of rapamycin mTOR inhibitors (mTORi) results in inevitable tumor regrowth, after the discontinuation of BRAFi, up to 75% of patients experienced rapid progression in a few weeks. However, upon rechallenge with BRAFi, 90% achieved an objective response [23].

Unfortunately, BRAFi are not free from the development of toxicity and adverse events (AE), and most of them have been reported in adult populations with melanoma, in which these new therapies are most commonly used, while few studies are available on pLGGs. Vemurafenib (a second-generation BRAFi) has already been studied in the treatment of BRAF V600E e V600K mutation-positive melanoma in the BRIM-3 Trial of 337 adult patients. In this trial, a total of 334 patients (99%) reported at least one AE in the vemurafenib arm. The most common AEs (occurring in >20% of patients) in the vemurafenib arm were rash, arthralgia, alopecia, fatigue, photosensitivity reaction, nausea, diarrhea, headache, hyperkeratosis, pruritus, dry skin, skin papilloma, decreased appetite, pain in extremity, pyrexia, vomiting, and squamous-cell carcinoma of the skin [24]. Preliminary data regarding BRAFi toxicities are also emerging in pediatric-age patients, and the most observed toxicities are pyrexia, hematological, dermatological, cardiac, and ophthalmic diseases [25]. The association of MEKi and BRAFi may mitigate some of the toxicities induced by the “paradoxical activation” of the MAPK pathway when a BRAFi is used as a single agent in BRAF wild-type cells [26,27]. Despite numerous studies that compare the different BRAFi molecules, there are no head-to-head clinical trials that compare the different agent combinations, and most of the safety data originate from the confirmatory phase III trials or pharmacovigilance studies in adult patients [24,28]. It is very important to study the SEs and the AEs in pediatric populations, considering the necessity of the long-term use of BRAFi (as is commonly carried out with mTORi in patients with tuberous sclerosis and SEGA). Moreover, the novel agents uncovered unexpected and unexplored AEs, and represent an important medical challenge. A better understanding of the SEs of these therapies is imperative.

The aim of our study is to expand the knowledge of the long-term AEs of BRAF inhibitors in pediatric populations by analyzing, retrospectively, the serum lipid concentrations in a cohort of pediatric patients treated with vemurafenib for LGG at our institute. We also discuss the possible role of lipid metabolism in resistance to BRAFi.

## 2. Materials and Methods

We collected and retrospectively reviewed clinical, laboratory, and instrumental data of all patients treated with the BRAF inhibitor vemurafenib for LGG at the Giannina Gaslini Children’s Hospital in Genoa, Italy, between 1 May 2015 and 31 December 2021. Patients treated with vemurafenib, with an age up to 18 years of age at the time of diagnosis, and with follow up of at least 6 months and 2 lipid-level samples (after starting treatment), were eligible.

The histology diagnosis was confirmed by the national reviewer, for all patients. All patients had a *BRAFV600E* mutation confirmed by sequencing performed using the polymerase chain reaction (kit Easy Braf real-time PCR, Diatech Pharmacogenetics, Jesi, Italy).

Before and during the BRAFi treatment (at 1, 3, 6, 12 months, and every 6 months thereafter), each patient underwent a detailed clinical and laboratory investigation to rule out possible organ disorders that contraindicated a BRAFi treatment (including a complete blood count, biochemical liver- and kidney-function tests, auxological and endocrinological assessments, ECG, and echocardiography with a cardiological examination). After the incidental finding of dyslipidemia in the second treated patient, fasting blood samples for lipid panel test were collected within 1 month before and at 1, 3, 6, 12 months after the initiation of treatment with vemurafenib (and every 12 months thereafter).

The auxological data were obtained from the auxo-endocrinological assessments to which the patients were routinely subjected. Height was measured by a Harpenden Stadiometer, with an accuracy of ±1 mm. The weight was measured on a digital scale, with an accuracy of ±0.1 kg. BMI was calculated as weight (kg) divided by height (m) squared and transformed to standard-deviation scores using the WHO reference values [29].

All the families receive a dietary recommendation based on the Mediterranean diet [30] at the first interview with the oncologist pediatrician. Patients with alterations in the lipid or glucose profiles are referred for nutritional counseling.

Fasting blood samples were obtained from 6 patients with LGG. Venous blood samples were collected by venipuncture or central venous catheter between 8 a.m. and 12 p.m., after an overnight fast. The serum and plasma were immediately separated, the lipid panel (triglycerides, total cholesterol, high-density lipoprotein cholesterol (HDL)) was quantified on the same day. An enzymatic colorimetric assay was used to determine total cholesterol, triglyceride, and direct HDL levels. Fasting plasma low-density lipoprotein cholesterol (LDL-C) was calculated using the Friedewald formula [31]. Fasting glycemia was determined using an enzymatic hexokinase assay.

Descriptive statistics were generated for the whole cohort, and data were expressed as mean and standard deviation for continuous variables. Median value and range were calculated and reported, as were absolute or relative frequencies for categorical variables. We analyzed the data available for all patients, before and after the incidental finding of dyslipidemia in the second patient (after which the systematic and scheduled analysis of the lipid profile was started). The box plots were used to show distributions of numeric variable values at 1 month before treatment, and at 1 month, 3 months, 6 months, 12 months, and at last follow-up after initiating treatment with vemurafenib. Box plots visually show the minimum value, the first quartile, the median, the third quartile, and the maximum value.

All data were analyzed with SPSS software for Windows (IBM SPSS Statistics for Windows, version 26. IBM Corp., Armonk, NY, USA).

This study was conducted in accordance with the Declaration of Helsinki. Informed consent was obtained from all the families.

## 3. Results

### 3.1. Study Population

We enrolled six patients (three males, three females) treated with the BRAFi vemurafenib, which was the first-choice BRAF target therapy in our hospital until December 2018. The demographic and clinical features of the six patients are reported in Table 1 and in Appendix A.

In the absence of pharmacokinetic and pharmacodynamic data for vemurafenib in pLGGs, we decided to start the treatment with a low dose of 370 mg/m^2^ (twice a day). Subsequent increases in the dosage were made every 2 weeks until the target range of 960–1100 mg/m^2^/day was reached after 1 month. Dose adjustments were made on a case-by-case basis during follow-up, depending on the patient’s drug tolerance. A patient with ganglioglioma reached the target dose after 30 months of starting the treatment, which was due to a resurgence of skin toxicity during dose-escalation attempts.

A patient with ganglioglioma was switched from the BRAFi vemurafenib to the BRAFi dabrafenib and the MEKi trametinib after 21 months of treatment because of severe skin toxicity. A patient with a pilocytic astrocytoma discontinued therapy with vemurafenib after 13 months because of tumor progression (neuroradiologically confirmed), which required antiedema therapy with dexamethasone, followed by chemotherapy and radiotherapy. In all the other patients, the treatment with BRAFi is still ongoing at the closing date of the database. The most frequent locations of tumors were the optic pathway/hypothalamic region (*n* = 4), followed by the basal ganglia (*n* = 1) and the spinal cord (*n* = 1). At the start of the target therapy, a tumor from one patient was disseminated.

The mean age at the start of targeted therapy was 8.4 ± 6.1 years (range: 3.5–18.8). The mean time from the last follow-up during treatment to the start of vemurafenib was 44.6 ± 26.5 months (range: 14.7–77.2). Before the vemurafenib, five out of six patients were treated with a neurosurgery partial resection, and five out of six with chemotherapy (one of these patients was also treated with radiotherapy) (Appendix A).

At the start of treatment, two patients were obese, and the mean BMI of the subjects included in the analysis was 0.9 ± 1.8 kg/m^2^. None of the subjects were taking any medication to specifically control glucose and/or lipid metabolism (such as statins, fibrate, hypoglycemic agents). They were free of overt liver, renal, and cardiac disease. The fasting blood glycemic levels were normal (Appendix A). The thyroid function and the hypothalamic–pituitary axis were normal or were well substituted with hormone-replacement therapies (Figure 1). The GH-deficiency treatment with rhGH in a patient was delayed by 12 months because of her oncological clinical condition.

### 3.2. Blood Lipid Levels

According to the 2011 National Heart, Lung, and Blood Institute (NHLBI) guideline [32], before treating patients with vemurafenib, the mean lipid levels of triglycerides (75.5 ± 24.9 mg/dL, *n* = 4), total cholesterol (157 ± 29.7 mg/dL, *n* = 3), and HDL (45.5 ± 4.9 mg/dL, *n* = 2;) were normal/acceptable, and the LDL levels were borderline (114 ± 14.1 mg/dL, *n* = 2).

One month after initiating treatment with vemurafenib, the lipid levels of triglycerides (107.8 ± 44.4 mg/dL, *n* = 4), total cholesterol (221.5 ± 42.1 mg/dL, *n* = 4), and LDL (139.5 ± 51.5 mg/dL, *n* = 4) were abnormal. The data remained elevated 3 months after the start of treatment: triglycerides (115 ± 45.6 mg/dL, *n* = 4), total cholesterol (238 ± 36.5 mg/dL, *n* = 4), and LDL (148.8 ± 40.2 mg/dL, *n* = 4). Moreover, the mean lipid levels of triglycerides (134.8± 83.6 mg/dL; Figure 2), total cholesterol (222.3 ± 34.7 mg/dL; Figure 3), and LDL (139.8 ± 46.9 mg/dL; Figure 4) were confirmed to be pathologically high at the last follow-up (from the start of treatment until the last follow-up, the mean time was 44.6 ± 26.5, range: 14.7–77.2 months). The HDL levels remained normal/acceptable (Figure 5).

The analysis of the individual patient data shows that the incidence of blood hypertriglyceridemia (according to the 2011 NHLBI) [32] after 1 month of vemurafenib was 50% (*n* = 2/4). According to the Common Terminology Criteria for Adverse Events v 5.0 (CTCAE) [33], one case was grade 0 and one case was grade 1, at long-distance follow-up, and the available incidence of hypertriglyceridemia was 84% after vemurafenib (*n* = 5/6, five cases were grade 0, and one case was grade 1 CTCA).

According to the 2011 NHLBI [32], hypercholesterolemia was presented in 100% of the patients (*n* = 4/4, all cases were grade 1 CTCA) after 1 and 3 months of vemurafenib. At long-distance follow-up, the available incidence of hypercholesterolemia after vemurafenib was 100% (*n* = 6/6, all cases were grade 1 CTCA). After 1 month, the LDL levels were elevated in 50% (*n* = 2/4), although there is no specific CTCA score for LDL, and all cases were grade 1 CTCA because of the required diet changes in the patients. Similarly, after 3 months, 75% (*n* = 3/, all cases were grade 1 CTCA), and, at long-distance follow-up, 83% (*n* = 5/6, all cases were grade 1 CTCA) of the vemurafenib group had elevated LDL levels.

## 4. Discussion

During the past years, target therapies have revolutionized therapeutic possibilities in oncology. The reports of side effects have increased proportionally to the increase in the number of available molecules. However, the knowledge of metabolic side effects is currently limited.

For the first time in children, we describe dyslipidemia (hypertriglyceridemia, hypercholesterolemia, and an increase in LDL) as common early and late adverse events after starting the BRAFi vemurafenib. However, the patients treated with vemurafenib were commonly pretreated with traditional chemotherapy. In a patient in whom vemurafenib was replaced by the combination of dabrafenib and trametinib for clinical reasons (photosensitivity and cutaneous side effects), the total cholesterol values returned to normal, as well as the other side effects, in a few weeks (Appendix A).

To the best of our knowledge, dyslipidemia, as an adverse event of BRAFi, is reported only in a few studies, and the blood-test screening of the occurrence is not included in the ongoing pediatric clinical trials. Therefore, a correlation between this possible side effect and resistance to BRAFi in clinical practice has not yet been explored. Because of the small cohort and design of our study, it was not possible to correlate dyslipidemia to the neuroradiological response after initiating BRAFi, and/or to resistance to the target therapy.

In a phase I study that investigated the pharmacokinetics, efficacy, and tolerability of vemurafenib (960 mg twice daily) in 42 Chinese patients (median age: 42, 19–69) with BRAFV600-mutation-positive unresectable or metastatic melanoma, dyslipidemia was a common AE, compared to the BRIM-3 study in Caucasians (cholesterol-level increase in 59% vs. <1%, hypertriglyceridemia in 22% vs. <1%) [24]. However, the blood-chemistry analysis of the full fasting lipid profile was not performed in the BRIM-3 protocol, and, therefore the incidence of dyslipidemia may be strongly underestimated. Severe hypercholesterolemia was reported (CTCA grade ≥ 3) in only 1 of 27 Chinese patients with hypercholesterolemia.

A significant increase in the plasma triglyceride levels was detected in 14 adult patients following vemurafenib treatment for V600E-mutated Erdheim–Chester disease (176.6 ± 22.2 vs. 130.7 ± 7.8 mg/dL untreated patients *n* = 42, +36%, *p* < 0.05) [34]. Interestingly, in the same study, although without statistically significant differences, the mean total cholesterol levels (214.5 ± 19.2 mg/dL) and the mean LDL levels (128.4 ± 20.6 mg/dL) were borderline-high (vs. acceptable mean values in non-treated V600E-mutated Erdheim–Chester disease: total cholesterol 174.5 ± 10.8 mg/dL, LDL 110.1 ± 9.6 mg/dL).

We know that some of the drugs that are used for targeted therapies have significant metabolic consequences, including dyslipidemia. On the other hand, the stimulation of lipid synthesis may result from the direct activation of oncogenic pathways in tumor cells. Many oncological mutations result in the aberrant activation of several signaling pathways, which can reprogram cancer-cell metabolism and cellular processes, including cell proliferation, differentiation, and the development of resistance to chemotherapy.

Among the target therapies, mTOR inhibitors are burdened with frequent dyslipidemia and, therefore, the etiopathology of this side effect has been the object of many studies. We know that the mTORi reduce the gene expression of lipogenic enzymes, such as acetyl-CoA carboxylase, fatty acid synthase, and stearoyl-CoA desaturase. Indeed, the mTORi are responsible for an increase in the total cholesterol and/or triglycerides by interfering with the protein kinase of the mTOR pathway [35].

New evidence supports that lipid metabolism is implicated in driving the tumor microenvironment and the cancer-cell phenotype, which contributes to the development and survival of cancer cells [36]. Changes in lipid metabolism can affect numerous cellular processes, including cell proliferation, differentiation, and motility [36]. In the tumor cells, the lipids can be used to store energy, synthesize the basic elements that are necessary for the cellular growth and proliferation (such as membranes), and participate in cell signaling [36,37]. Cancer cells compete for oxygen and nutrients with the host cells, and they maintain their malignant potential by modifying the lipid metabolism. The oxidative catabolism of lipids provides ATP and NADH, both of which are essential to controlling environmental stress and promoting survival [37,38].

Therefore, lipid-metabolism reprogramming is an essential link between the tumor and the host metabolism, with implications in sensitivity to chemotherapies [37], including target therapies [39].

A potential link between BRAFV600E and lipid-metabolism regulation in cancer cells is suggested by some cell and mouse model studies [37,40,41]. In 2015, Kang et al. [42] demonstrated the interaction between oncogenic BRAF V600E and the enzyme 3-hydroxy-3-methylglutaryl-CoA lyase (HMGCL), which is involved in lipid metabolism by producing ketone bodies. HMGCL expression is upregulated in BRAF V600E melanoma and hairy-cell leukemia. BRAF upregulates HMGCL through an octamer transcription factor, Oct-1, which leads to increased intracellular levels of the HMGCL product, acetoacetate, which selectively enhances the binding of the BRAF V600E, but not the BRAF wild-type to MEK1 in V600E-positive cancer cells, to promote the activation of MEK–ERK signaling and, therefore, tumor growth. In 2017, Xia et al. [43] showed that a high-fat ketogenic diet increased the serum levels of acetoacetate, which led to the potential tumor growth of BRAF V600E-expressing human melanoma cells in xenograft mice. The high-fat diets resulted in increased growth rates, masses, and sizes of tumors, without affecting the body weight in these mice. In contrast, a high-fat diet did not affect the tumor growth rates, masses, sizes, or the body weight in mice with tumor xenografts expressing an active NRAS Q61R mutation. The increased tumor growth in xenograft mice (BRAF mutated) fed with a high-fat diet was not due to differences in the quantity of the food intake. In both mice models, the consumption of a high-fat diet did not significantly affect the serum levels of D-b-hydroxybutyrate (3HB), but significantly increased the serum cholesterol levels compared to control mice fed with a normal diet. Treatment with hypolipidemic agents or an inhibitory homolog of acetoacetate attenuated the BRAF V600E tumor growth [43].

Valvo et al., in 2021 [41], showed that, in BRAFV600E papillary thyroid carcinoma, the de novo lipid synthesis significantly increased (1.58- and 1.34-fold changes in heterozygous and homozygous BRAFV600E -derived cell lines, respectively) within 6h in vemurafenib-treated cancer cells. The xenograft mouse data further showed that human BRAFV600E tumor cells became less responsive to vemurafenib within two weeks, and ultimately exhibited increased tumor growth when the *Acetyl-CoA Carboxylase 2 gene* (*ACC2*) was knocked down. This suggests that silencing the ACC2 (a rate-limiting enzyme for de novo lipid synthesis and the inhibition of fatty acid oxidation) may contribute to BRAFV600E-inhibitor (e.g., vemurafenib) resistance and increased tumor growth. BRAFV600E inhibition increased the de novo lipid-synthesis rates, decreased fatty acid oxidation due to the oxygen-consumption rate, and increased the intracellular reactive-oxygen-species (ROS) production, which can trigger tumor-cell proliferation or death [41].

The modulation of numerous genes, including multiple oncogenes, growth factors, and tumor suppressors, are activated by reactive oxygen species (ROS) and the modification of the level of the AMP/ATP ratio that is due to cancer-cell metabolic plasticity (both possible effects of cancer and anticancer therapy, including BRAFi and MEKi) [44]. The HIF-1 and AMP-activated protein kinase (AMPK), which operate as energy biosensors of oxidative stress and master regulators of cellular metabolism, play a crucial role in this phenomenon. [45,46]. The AMPK regulates the ATP level through the switch from anabolic to catabolic metabolism via the stimulation of glucose uptake, aerobic glycolysis, and mitochondrial oxidative metabolism, which is mainly due to the β-oxidation of fatty acids [46]. These pathways interplay with HIF-1, and, therefore, a variety of oncogenes, such as *Ras*, *c-Myc*, and *p53*, and the Akt/PKB, PI3K, and mTOR signaling pathways, sustain cancer-cell proliferation and survival [47,48,49]. The gene *KRAS* is also directly implicated in ROS generation by NADPH oxidases [50]. Cancer-cell survival and metastasis can be sustained by lipid biosynthesis that is promoted by a shift in the glutamine metabolism from oxidation to reductive carboxylation [49].

In BRAF V600E melanoma cells, altered lipid metabolism could contribute to targeted therapy resistance through the modification of the activation of several lipogenesis pathways [51,52,53]. New evidence shows that the SREBP-1-dependent activation of lipogenesis is required for tumor growth and for cell survival in multiple cancer models, including high-grade glioma [54,55]. In BRAF-mutant melanomas, therapy resistance to vemurafenib is supported by the Sterol Regulatory Element-Binding Protein (SREBP1) activation [56] and the upregulation of the S1 P-dependent signaling pathway [37,40,51,52,57]. In sensitive BRAF-mutant models, vemurafenib caused the decrease in lipogenesis and the activation of SREBP-1. All showed high levels of lipogenesis, even in the presence of the inhibitor. However, this was not seen in therapy-resistant models, in which BRAFi only induced a moderate decrease in the SREBP-1 levels and did not significantly affect lipogenesis [56]. Probably this is due to the activation of the alternative ERK pathway that is linked to therapy resistance and that is a known regulator of SREBP [58,59], as is shown by the decreased levels of SREBP-1 in therapy-resistant cells treated with the MEK inhibitor trametinib [56]. The expressions of well-established mSREBP-1 downstream targets, such as ACLY, ACACA, and FASN, were also consistently reduced. These findings indicate that the reactivation of the ERK pathway contributes to sustained SREBP-1 activity in therapy-resistant melanoma cells [56]. Moreover, the expression of key lipogenic enzymes—SREBP-1 downstream targets—such as fatty acid synthase (FASN) acetyl-CoA carboxylase-1, were found to be inversely associated with drug resistance in BRAF-mutant cell lines [56].

In the current study, most of these laboratory AEs met the criteria as AEs because the events were medically significant and required diet modification. All these events were grade 1 CTCA, asymptomatic, and did not require a change in treatment or dose modification. However, because of the possible need for long-term use, these observed results may affect the overall benefit/risk assessment of vemurafenib in patients with high cardiovascular risk.

## 5. Conclusions

The targeted therapies for brain tumors are innovative and promising oncological treatments, and as a result, their use has expanded widely. The effectiveness of BRAFi, and its use in combination with other new target therapies, is increasing, and therefore the spectrum of side effects needs to be further explored.

The toxicities that are related to these new agents are generally not life threatening; however, the long-term effects are unknown, and they could potentially be a limiting factor in chronic life-long use. An accurate screening strategy in new clinical trials, and a multidisciplinary team, are required for the optimal management of unexpected adverse events.

We describe, for the first time, the possible side effects of BRAFi in a case series of children treated for LGG. In our study, children treated with vemurafenib showed a worsening in their lipid profiles, with a significant increase in triglycerides, LDL, and total cholesterol over time. New prospective and multicentric clinical trials of larger study groups are needed to confirm our observation; therefore, the evaluation of the serum lipid balance should be implemented in future experimental protocols, including BRAFi and/or MEKi.

Because of the large amount of data that show the possible role of lipid metabolism in the mechanisms of resistance and response to biological therapies, new future studies should explore this hypothesis.

## Figures and Tables

**Figure 1 cancers-14-02693-f001:**
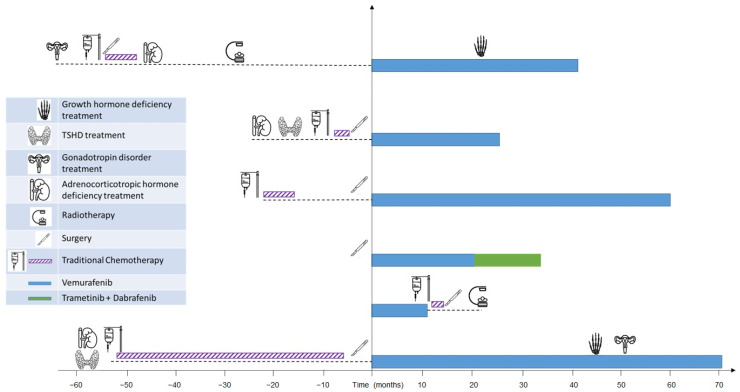
Swimmer plot, endocrinological and oncological treatments.

**Figure 2 cancers-14-02693-f002:**
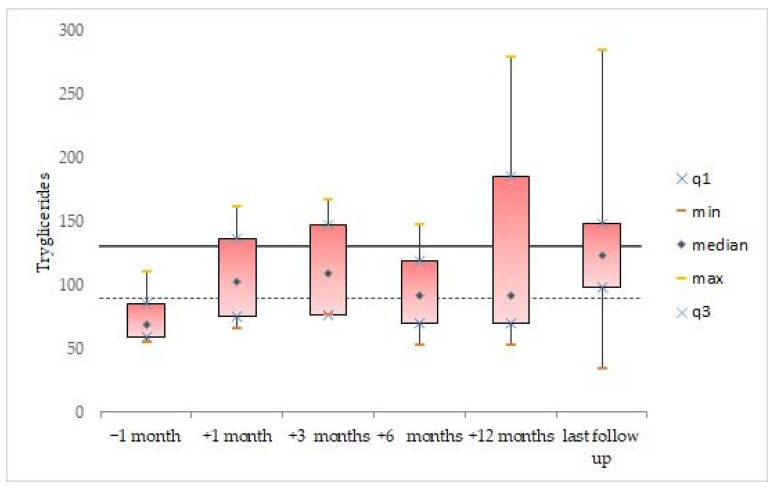
Triglyceride levels before and during vemurafenib. The dashed line indicates the acceptable upper limit, the double line indicates the borderline-high limit according to the NCEP Expert Panel on Cholesterol Levels in Children.

**Figure 3 cancers-14-02693-f003:**
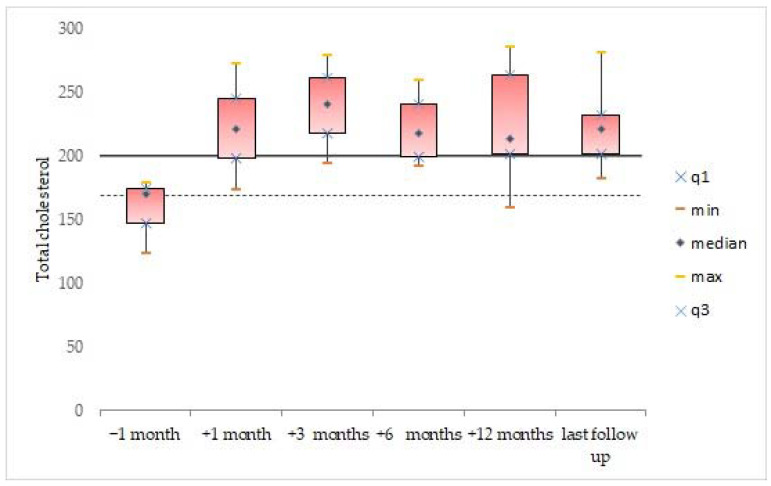
Total cholesterol levels before and during vemurafenib. The dashed line indicates the acceptable upper limit, the double line indicates the borderline-high limit according to the NCEP Expert Panel on Cholesterol Levels in Children.

**Figure 4 cancers-14-02693-f004:**
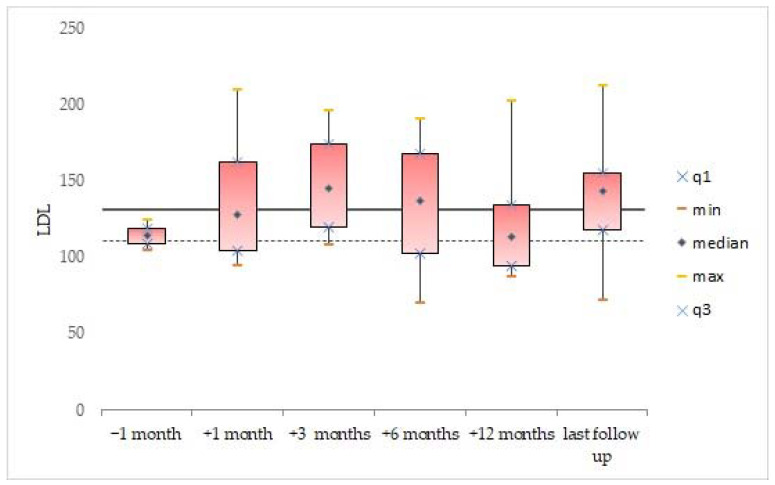
LDL levels before and during vemurafenib. The dashed line indicates the acceptable upper limit, the double line indicates the borderline-high limit according to the NCEP Expert Panel on Cholesterol Levels in Children.

**Figure 5 cancers-14-02693-f005:**
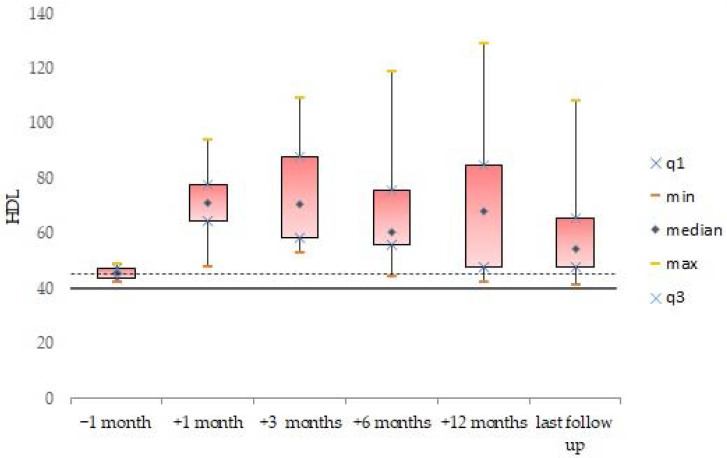
HDL levels before and during vemurafenib. The dashed line indicates the acceptable upper limit, the double line indicates the borderline-high limit according to the NCEP Expert Panel on Cholesterol Levels in Children.

**Table 1 cancers-14-02693-t001:** Demographic and clinical features of the patients treated with vemurafenib.

Characteristic	Vemurafenib Group (*n* = 6)
**Sex number:**	
male/female	3/3
**Age at diagnosis:**	
mean years ± SD	5.5 ± 5.6
median	3.4 (0.3; 13.8)
**Age at the start of vemurafenib:**	
mean years ± SD	8.4 ± 6.1
median	7.1 (2.8; 18.8)
**Tumor site:**	
hypothalamic/chiasmatic	4
basal ganglia	1
spinal cord	1
**Tumor histology:**	
ganglioglioma	4
pilocytic astrocytoma	2
**Previous treatment:**	
surgery or biopsy only	1
one line of chemotherapy	2
two lines of chemotherapy	1
≥three lines of chemotherapy	1
≥three lines of chemotherapy + RT	1

## Data Availability

The data presented in this study are available on request from the corresponding author.

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
