# Peer review of "Dyslipidemia in Children Treated with a BRAF Inhibitor for Low-Grade Gliomas: A New Side Effect?"

_cancers, 2022, doi:10.3390/cancers14112693_

Round 1

Reviewer 1 Report

The authors focused on an aspect of high clinical significance in an era of informed clinical decisions, accounting also for the fact that most clinical trials still refer to adult populations (very limited data are available for paediatric or geriatric populations, same for pregnant women, etc). The study has two objectives, one being the long term AEs of BRAFi (in which vemurafenib serves as a paradigm) in children and teenagers and the second being the possible role of lipid metabolism in BRAFi resistance. Overall this is a well written manuscript with a clear rationale also commenting on the limitations of the study. Some minor concerns are shown bellow:

The authors state "Descriptive statistics were generated for the whole cohort and data were expressed 139 as mean and standard deviation for continuous variables. Median value and range were 140 calculated and reported, as were absolute or relative frequencies for categorical variables". Please clarify that the analysis refers to both retrospective and prospective individuals. Please include (supplementary table?) the demographic and clinical features for the retrospective individuals taken into account (i.e. similarly to Table 1) so that the readership can appreciate the outcomes.

Please elaborate on your statistical methods to be of benefit to the readership.

Figure1 legend appears upside-down, please correct

Reviewer 2 Report

In this article, Crocco M. et al presented and discussed the risk of dyslipidemia due to utilization of BRAF inhibitors vemurafenib on pediatric low-grade glioma (pLGG) patients. The authors noted the alterations in few lipid metabolism markers: TC, TG & HDL observed 1 month into the treatment and which persisted few years down. 

As the authors mentioned, considering the great survival rate of pLGG, it is crucial to consider carefully onco-drugs which may exert AE, including the case of dyslipidemia.

Overall, although the article is comparatively short, the authors presented crucial information supported with quite thorough introduction & discussion. I believe this article is of significant quality and would attract the interest of Cancers' community. Thus, I recommend the acceptance of this article following adequate clarifications below: 

  • Figure 1 is mirrored/rotated. Please ensure it is corrected as it provides crucial background to the cohort subjects.
  • I could not find dosage information related to vemurafenib administration. Is there any variation to amount administered to the 6 subjects? 
  • The authors mentioned in the discussion part that a patient was subsequently given dabrafenib and trametinib to replace the vemurafenib due to clinical reasons, and the TC and other side effects returned normal. Further elaboration on this subject would be helpful (e.g. rate by which the TC/other lipid side effects returned) 

Reviewer 3 Report

In this manuscript, the authors reported dyslipidemia as a possible side effect of vemurafenib, a BRAF inhibitor. This manuscript is recommended to be published in this journal after supplying with additional data and discussion. 

  1. Please provide the fasting blood glucose level before and during treatment.
  2. Due to that lipid metabolism is mainly performed in liver, please provide the biochemical liver and kidney function test.
  3. Will the MEKi (trametinib) cause dyslipidemia?

  4. Will the dyslipidemia recover after the treatment or how long will the dyslipidermia become normal?
